# Structure and nature of ice XIX

Christoph G. Salzmann [1✉], John S. Loveday[2], Alexander Rosu-Finsen [1] & Craig L. Bull [3]

Ice is a material of fundamental importance for a wide range of scientific disciplines including physics, chemistry, and biology, as well as space and materials science. A well-known feature of its phase diagram is that high-temperature phases of ice with orientational disorder of the hydrogen-bonded water molecules undergo phase transitions to their ordered counterparts upon cooling. Here, we present an example where this trend is broken. Instead, hydrochloric-acid-doped ice VI undergoes an alternative type of phase transition upon cooling at high pressure as the orientationally disordered ice remains disordered but undergoes structural distortions. As seen with in-situ neutron diffraction, the resulting phase of ice, ice XIX, forms through a *Pbcn*-type distortion which includes the tilting and squishing of hexameric clusters. This type of phase transition may provide an explanation for previously observed ferroelectric signatures in dielectric spectroscopy of ice VI and could be relevant for other icy materials.

[1] Department of Chemistry, University College London, London, UK. [2] School of Physics and Astronomy and Centre for Science at Extreme Conditions, University of Edinburgh, Edinburgh, UK. [3] ISIS Neutron and Muon Facility, Rutherford Appleton Laboratory, Didcot, UK. ✉email: c.salzmann@ucl.ac.uk

The phase diagram of water displays tremendous complexity with currently eighteen known ice polymorphs, a highly anomalous liquid phase and at least two families of amorphous ices[1–3]. The $H_2O$ molecule is a versatile molecular building block capable of forming at least eleven different types of four-fold connected crystalline networks through hydrogen bonding[1]. For a given network topology, the hydrogen-bonded water molecules can either display orientational disorder or order resulting in so-called hydrogen-disordered and hydrogen-ordered phases of ice, respectively. The general trend is that hydrogen-disordered phases of ice crystallise from the liquid phase which are then expected to undergo hydrogen-ordering phase transitions at low temperatures to their hydrogen-ordered counterparts in line with the third law of thermodynamics (see Fig. 1a). However, due to the highly cooperative nature of the molecular reorientation processes in ice, the hydrogen disorder is often frozen in upon cooling resulting in orientational glasses with glass transition temperatures above the hydrogen-ordering temperatures[4]. Acid and base dopants, such as hydrohalic acids or alkali hydroxides, can speed-up the molecular reorientations in ice through mobile point defects which lowers the glass transition temperature. Effective dopants lower the glass transition temperature below the hydrogen-ordering temperature which means that hydrogen-ordering can take place[5–7]. The hydrogen-ordered phases of ice are typically antiferroelectric which means that the crystals do not display macroscopic dipole moments. The one notable exception discovered so far is the ferroelectric ice XI which is the hydrogen-ordered counterpart of the 'ordinary' ice Ih[8].

Ice VI is a hydrogen-disordered phase of ice that crystallises from liquid water in the 0.6–2.2 GPa pressure range[9]. Its structure consists of two interlocking hydrogen-bonded networks and therefore it is often described as a self-clathrate (see Fig. 1b)[9]. The networks themselves are built of hexameric clusters that are hydrogen bonded to one another in the $a$ and $b$ crystallographic directions and share edges along the $c$ direction (see Fig. 1c). These clusters contain four waist and two apex molecules and their structure is the same as the cage-like $(H_2O)_6$ clusters in the gas phase[10,11]. Upon cooling pure ice VI, the hydrogen disorder is frozen-in as observed by neutron diffraction[12]. However, dielectric spectroscopy measurements have indicated a very slow transition to a ferroelectric state below ~125 K[13,14]. This discrepancy has so far not been clarified. A break-through in hydrogen ordering ice VI came with using hydrochloric acid (HCl) as a dopant which led to the discovery of the antiferro-electric hydrogen-ordered ice XV[6]. The transition from ice VI to ice XV is accompanied by an increase in volume which means that the most ordered ice XV samples were obtained upon slow-cooling at ambient pressure[7,15]. Upon cooling HCl-doped ice VI at pressures above ~1 GPa, the hydrogen-ordering phase transition from ice VI to ice XV is suppressed progressively even though the glass transition temperature of the molecular reorientation dynamics has been lowered significantly by the acid dopant[6,15–18]. These unusual doped samples have been called deep-glassy ice VI[19]. Upon heating at ambient pressure, deep-glassy ice VI shows the phenomenon of transient ordering as the metastable ice VI state first undergoes irreversible hydrogen ordering to ice XV and then reversible hydrogen disordering to ice VI[15,17].

Recently, it was shown that HCl-doped ice VI samples cooled at pressures above 1.4 GPa display an endothermic feature before the exothermic transient ordering upon heating at ambient pressure and at certain heating rates[19,20]. As proposed by the ice $\beta$-XV scenario, the low-temperature endotherm arises from a new hydrogen-ordered phase of ice that is more hydrogen-ordered, differently hydrogen-ordered and more stable than ice XV[20,21]. In contrast to this, it has been pointed out that the new endotherm is consistent with a kinetic overshoot effect associated with the underlying glass transition of the deep glassy ice VI[19]. Upon increasing the pressure, more relaxed states are obtained that give more pronounced endotherms. The kinetic origin of the endotherms was illustrated very clearly by varying the heating rate: Depending on the state of relaxation, fast heating could make the endotherm appear whereas slow-heating leads to its disappearance[19]. It has also been argued that the X-ray diffraction and dielectric spectroscopy data in ref. [20] are consistent with the deep glassy ice VI scenario[19]. Following a Raman spectroscopic study[21], combined inelastic neutron spectroscopy and neutron diffraction data of an HCl-doped ice VI slow-cooled at 1.7 GPa were presented[22]. Both the spectroscopic and the diffraction data showed that the sample was structurally very similar to standard pure ice VI. The deep glassy ice VI scenario requires the samples to decrease slightly in entropy as the pressure is increased. It was speculated that local and orientationally uncorrelated hydrogen ordering may be occurring as the pressure is increased[22]. This could lead to increased levels of local distortions that could eventually even reduce the space group symmetry of ice VI[22].

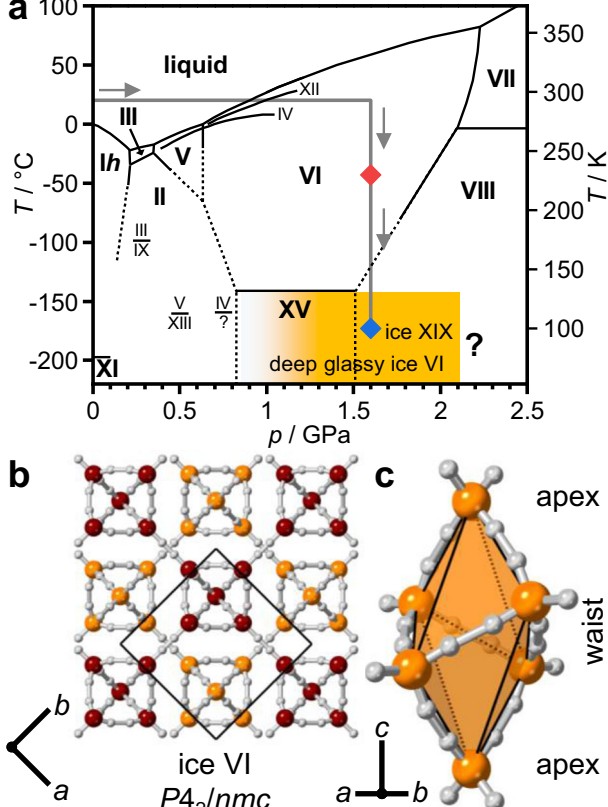

**Fig. 1 Phase diagram of ice[1] and crystal structure of ice VI[9]. a** The experimental pathway of the neutron diffraction experiment is highlighted by a thick grey line and the stable phases are shown with bold fonts. Solid and dashed black lines indicate experimentally measured and extrapolated phase boundaries, respectively. High-quality diffraction data were collected at 230 and 100 K at ~1.6 GPa as indicated by the red and blue diamonds. The formation of deep glassy ice VI starts gradually above 1.0 GPa as indicated by the orange-shaded area. **b** The hydrogen atoms of the ice VI structure are shown in white, and the oxygen atoms of the two individual networks in burgundy and orange, respectively. The unit cell is indicated by a black square and the crystallographic axes are defined below the crystal structure. **c** Hexameric unit which is the basic building block of an ice VI network.

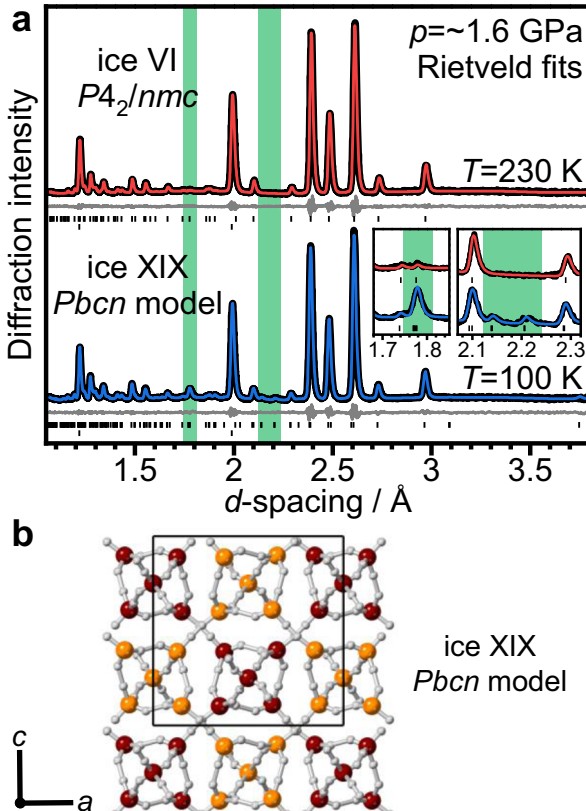

**Fig. 2 Analysis of the diffraction data of DCl-doped D₂O ice VI samples collected at 1.6 GPa. a** Rietveld fits of the diffraction data collected 230 and 100 K using the $P4_2/nmc$ space group for ice VI and $Pbcn$ for ice XIX. The upper tick marks indicate the expected positions of Bragg peaks for the ice whereas the lower tick marks are from the sintered diamond anvil. The insets show the fits in the regions where the most major peak intensity change and the additional weak Bragg peaks are observed for ice XIX. **b** Crystal structure of ice XIX using the $Pbcn$ space group. The hydrogen atoms are shown in white, and the oxygen atoms of the two individual networks in burgundy and orange, respectively.

In this work, using in-situ pressure powder neutron diffraction we find evidence for such distortion phenomena of the hydrogen-disordered structure and a systematic analysis of the symmetry changes is presented. The mechanically most meaningful type of distortion is also the one that provides the best fit to the diffraction data. As a final point, we highlight similarities between the pressure-induced ice VI distortion and those observed in perovskite materials.

## Results

**In-situ neutron diffraction.** Figure 2a shows the neutron diffraction patterns of a D₂O ice sample doped with 0.01 molar DCl at ~1.6 GPa. The diffraction pattern at 230 K is consistent with fully hydrogen-disordered ice VI with $P4_2/nmc$ space group symmetry. Remarkably, slow-cooling to 100 K at ~0.5 K min⁻¹ and ~1.6 GPa leads to the appearance of two very weak additional Bragg features at ~2.14 and ~2.21 Å, as shown in Fig. 2a. Another significant crystallographic change is the increase in intensity of the Bragg feature at ~1.77 Å. Both changes are highlighted by green-shaded regions in Fig. 2a. As shown in Supplementary Fig. 1, the $P4_2/nmc$ ice VI model fits the low-temperature diffraction data reasonably well. However, the additional peaks and the intensity increase at ~1.77 Å cannot be reproduced. This means that a new phase of ice has formed that is structurally

closely related to ice VI but with a lower space group symmetry. The tickmarks of a corresponding $P1$ structure with the lattice constants of ice VI in Supplementary Fig. 1 illustrate that the size of the unit cell must be increased to produce the additional Bragg peaks. Indexing of the diffraction pattern suggests an increase in the size of the unit cell to a $\sqrt{2} \times \sqrt{2} \times 1$ supercell. Compared to the ice VI unit cell, which comprises two hexameric clusters, the $\sqrt{2} \times \sqrt{2} \times 1$ supercell contains four hexameric clusters.

**Structural distortions and space group symmetry.** In Supplementary Table 1, we present a systematic and general analysis of the various possible crystallographic subgroups of the $P4_2/nmc$ space group of ice VI. This shows that the increase of the unit cell to a $\sqrt{2} \times \sqrt{2} \times 1$ supercell is quite frequently encountered. Here, a more intuitive crystal-chemical approach is presented that considers local distortions of the hexameric units as the origin for the change in space group symmetry in line with the conclusions of our spectroscopic study[22]. As it turns out, a $\sqrt{2} \times \sqrt{2} \times 1$ supercell is the smallest cell needed for describing meaningful distortions of the individual networks. Even the visual appearance of the ice VI structure with its linked hexameric clusters suggests that such a structure could be prone to local distortions (see Fig. 1a).

Figure 3 shows the various ways in which the ice VI structure can distort based on the $\sqrt{2} \times \sqrt{2} \times 1$ supercell. Firstly, all permutations of tilting the hexameric clusters with respect to each other is considered which leads to a reduction of the space group symmetry from $P4_2/nmc$ to $Pbcn$, $P2/c$, and $Ccc2$. Out of these, only $Pbcn$ allows reflections at the positions of the additional Bragg features. Furthermore, the $Pbcn$ distortion is the only one that makes mechanical sense as the neighbouring hydrogen-bonded clusters always tilt in the opposite direction as shown schematically in Fig. 3b. Shearing of the hexameric clusters is consistent with the $Pbcn$ and $P2/c$ space groups. The $Pbcn$ distortion seems to again be more favourable since the molecules at the interfaces between the two networks always move in the same direction. The complete displacement of the two neighbouring networks with respect to each other is allowed in the $Pnna$ space group. A final way to distort the hexameric clusters are squishing modes where they are compressed in one direction and expanded in the perpendicular direction. Going through all permutations, the resulting space groups are $Pbcn$, $P2/c$, and $Ccc2$ of which $Pbcn$ is again the only one consistent with the additional Bragg features. Considering that the distortions take place at high pressure, the $P2/c$ and $Ccc2$ models are also mechanically unlikely since they would require long-range expansion in one or even two dimensions, respectively. The $Pbcn$ squishing modes on the other hand do not require a volume change since neighbouring hydrogen-bonded clusters always squish in the opposite direction. In summary, $Pbcn$ allows a multitude of mechanically meaningful distortions of the ice VI structure and permits the additional Bragg features seen in the diffraction pattern at 100 K and ~1.6 GPa. The $Pnna$ shearing distortion also allows the new Bragg peaks and will therefore need to be investigated as well.

Figure 3b illustrates nicely how the additional Bragg features appear in calculated diffraction patterns as the ice VI structure is distorted by tilting the hexameric clusters by the angle $\alpha$ according to $Pbcn$. Remarkably, not only do the additional Bragg features appear and increase in intensity with $\alpha$, the distortion also increases the intensity of the Bragg peak at ~1.77 Å which we observed earlier in the experimental diffraction pattern collected at 100 K (see Fig. 2a).

In a next step, the fully hydrogen-disordered $Pbcn$ model was refined against the experimental low-temperature data using the Rietveld method. As can be seen in Fig. 2a, a very good match with respect to the intensities of all Bragg peaks in the

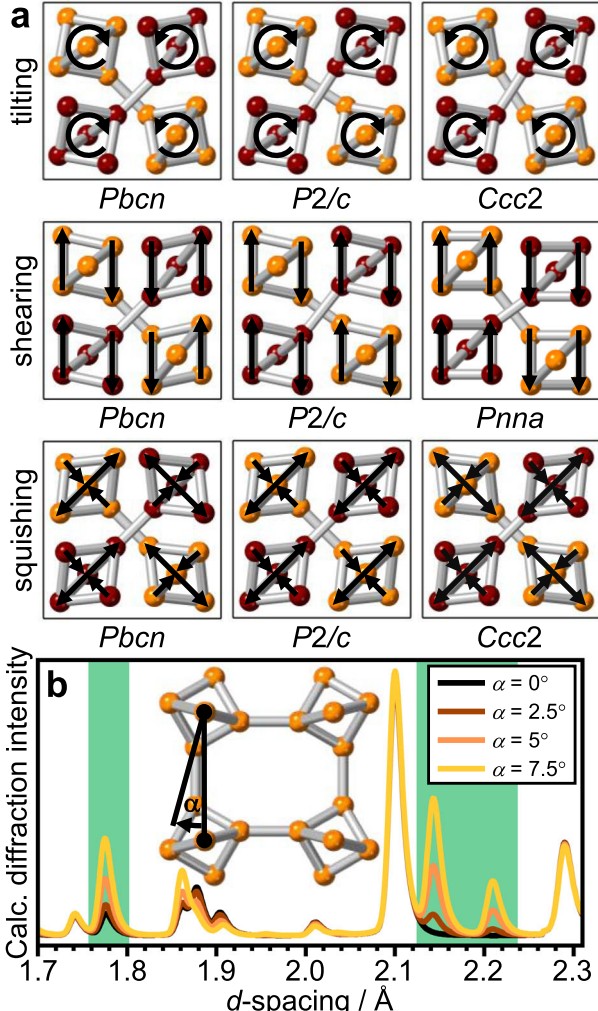

**Fig. 3 Distortions of the ice VI structure and their effects on diffraction. a** Schematic illustration of the possible distortions of ice VI by tilting, sheering, and squishing the hexameric units as illustrated by the arrows. The oxygen atoms of the two individual networks are shown in burgundy and orange, respectively. **b** Calculated diffraction patterns illustrating the emergence of ice XIX diffraction characteristics as a function of the tilt angle $\alpha$ for a *Pbcn* distortion. The green-shaded areas correspond to the ones shown in Fig. 2a. As shown in the inset, the axis of rotation is perpendicular to the paper plane and the angle is calculated from the projections into the plane of both the vector between the apex oxygens of neighbouring clusters and the apex to waist oxygen vector within a given cluster.

distortions break the tetragonal $P4_2/nmc$ symmetry of ice VI to the orthorhombic *Pbcn*. The $a$ and $c$ lattice constants of the *Pbcn* structure take slightly different values of $8.3966 \pm 0.0003$ Å and $8.3737 \pm 0.0003$ Å, respectively, reflecting the orthorhombic splitting. The full crystallographic information of the *Pbcn* structure is given in Table 1.

Reassuringly, the systematic subgroup analysis shown in Supplementary Table 1 delivered all the possible space groups that also resulted by considering the various types of distortions. The six possible candidate structures from the subgroup analysis are shown in Supplementary Fig. 2. Supplementary Fig. 3 shows that out of these, the *Pcnb* structure, which is equivalent to the *Pbcn* structure discussed earlier, gives the best possible fit to the diffraction data. Based on this analysis, the previously discussed *Pnna* model can also be discarded. Accordingly, both the systematic subgroup analysis, as well as the crystal-chemical approach considering the various types of distortions, suggest the *Pbcn* space group.

**Bottom-up search for space group symmetry.** Even though the *Pbcn* model gives a very good fit to the diffraction data, it is worth considering if the actual symmetry of the distorted structure is a subgroup of *Pbcn* which is always a possibility in crystallography. The apex oxygen atoms lie on a special Wyckoff position in *Pbcn* which means that the two networks are not permitted to shift against each other in $b$ direction. Allowing such shifts would result in a reduction of the space group symmetry from *Pbcn* to $P2/c$ (with a different axis compared to the $P2/c$ model shown in Fig. 3a). Additional shifts in either $a$ or $c$ directions or both take the space group symmetries down to $Pc$ and $P\bar{1}$, respectively. Considering that the hexameric clusters are hydrogen disordered and hence have a multitude of different structures with molecules in different orientations, it seems possible that a wide range of asymmetric local distortions take place as the clusters are distorted. Considering this asymmetry, the resulting space group symmetry would be $P1$.

Instead of analysing the low-symmetry crystallographic models with the Rietveld method, we take a pragmatic approach in the next step and fit the diffraction data with a Reverse Monte Carlo (RMC) approach using the RMCProfile software[23]. Here, a large supercell without any symmetry and 1500 water molecules is used. The fit to the experimental diffraction data is achieved by the movements of the 4500 individual atoms. Since diffraction data is fitted, the resulting structure after fitting will be consistent with the average structure of the sample. To be able to extract local information from the simulation box in the absence of total scattering data, we implemented a simple computational model with harmonic potentials between the oxygen and hydrogen sites and between the two hydrogen sites along one hydrogen bond. One of the hydrogen sites along a given hydrogen bond is occupied and the other empty as required by the ice rules, and the overall structure was ensured to be fully hydrogen disordered using our RandomIce program[15]. In addition to implementing a meaningful local structure, the potentials also ensure that the connectivity of the hydrogen-bonded network remains unchanged and that the structure stays hydrogen disordered. RMCProfile then attempts to fit the diffraction data while simultaneously minimising the total energy of the supercell[23].

The experimental diffraction data could be easily fitted with the RMC approach, as shown in Fig. 4a including the additional Bragg features and the increased intensity of the ~1.77 Å peak in the low-temperature data. The pair distribution functions, $g(r)$, shown in Fig. 4b reflect the probabilities of finding atoms at certain distances away from a central atom. Overall, the O-D and O-O $g(r)$ functions look quite similar for the 230 and 100 K data

experimental data is obtained including the ~1.77 Å and the additional weak Bragg peaks ($wRp = 0.0206$, $Rp = 0.0205$).

The crystal structure resulting from the converged Rietveld refinement of the *Pbcn* model clearly shows distortions of the hexameric clusters (see Fig. 2b). The tilting angle, as defined in Fig. 3b, is about 3°, which leads to long-range snake-like distortions upon moving from one cluster to the next within the same network. The shearing displacement is minor with 0.07 Å. The squishing distortions, however, are significant. The distances between the waist oxygen atoms across the hexameric clusters are 3.15 and 3.53 Å, respectively. For comparison, in ice VI at 230 K, these distances take equal values of 3.31 Å. Interestingly, the squishing of the clusters leads to slight twisting of the hydrogen bonds leading from the waist to the apex molecules which seems a meaningful mechanical response to the squishing distortion. Overall, the

**Table 1 Fractional atomic coordinates, fractional occupancies, order parameters, and isotropic atomic-displacement parameters ($U_{iso}$) of DCl-doped $D_2O$ ice XIX at 100 K and ~1.6 GPa and using the _Pbcn_ structural model.**

| Atom label | Atom type | x | y | z | Occupancy | $U_{iso}*100$ |
|---|---|---|---|---|---|---|
| D1 | D | 0.2428(9) | 0.1142(11) | 0.0299(6) | 0.5 | 1.199(32) |
| D2 | D | 0.8007(7) | 0.3621(12) | 0.0305(7) | 0.5 | 1.199(32) |
| D3 | D | 0.0673(7) | 0.2313(11) | 0.1053(8) | 0.5 | 1.199(32) |
| D4 | D | −0.0383(9) | 0.3205(11) | 0.0765(8) | 0.5 | 1.199(32) |
| D5 | D | 0.0849(7) | −0.0326(11) | 0.1278(8) | 0.5 | 1.199(32) |
| D6 | D | −0.1025(10) | 0.5366(11) | 0.1483(9) | 0.5 | 1.199(32) |
| D7 | D | 0.0454(9) | 0.8920(10) | 0.1885(7) | 0.5 | 1.199(32) |
| D8 | D | −0.0717(7) | 0.6720(10) | 0.2079(7) | 0.5 | 1.199(32) |
| D9 | D | 0.1860(9) | 0.2308(12) | 0.1868(7) | 0.5 | 1.199(32) |
| D10 | D | 0.8424(13) | 0.2873(13) | 0.2072(7) | 0.5 | 1.199(32) |
| O1 | O | 0.1502(5) | 0.1182(6) | 0.1022(5) | 1 | 0.657(33) |
| O2 | O | 0.8737(4) | 0.3726(6) | 0.1109(6) | 1 | 0.657(33) |
| O3 | O | 0 | 0.7349(12) | 0.25 | 1 | 1.421(33) |

The lattice constants are: $a = 8.3966(3)$ Å, $b = 5.4546(1)$ Å, and $c = 8.3737(3)$ Å. Numbers in parentheses are statistical errors of the last significant digit of refined quantities.

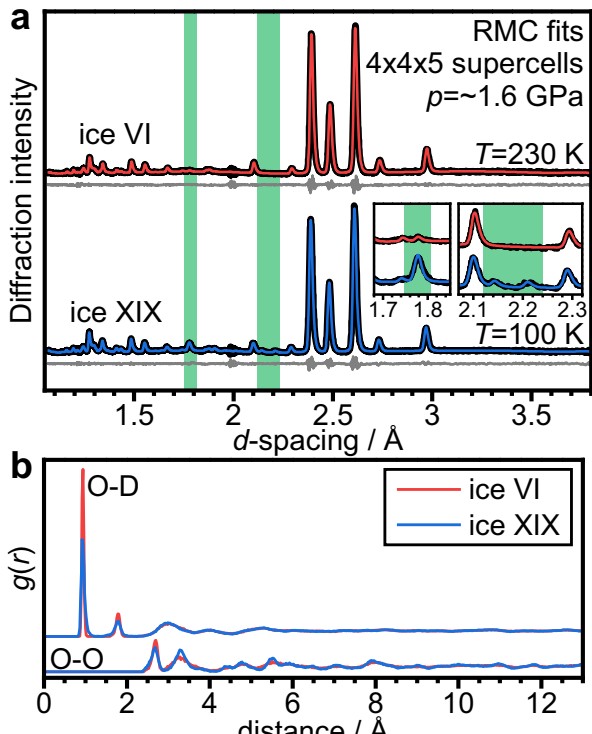

**Fig. 4 Symmetry-free fitting of the ice VI and ice XIX diffraction data with a Reverse Monte Carlo approach (RMCProfile) and a simple spring model to describe intra- and intermolecular bonding. a** Fit to the diffraction data using hydrogen-disordered ice VI supercells containing 1500 $D_2O$ molecules. The green-shaded areas correspond to those shown in Fig. 2a and highlight the diffraction ranges where the most major intensity increase and the new Bragg peaks characteristic for ice XIX were observed. **b** Resulting O-D and O-O pair-distributions functions, $g(r)$, for the two structures.

without local and asymmetric distortions would lead to sharpening of these features due to the reduction in the thermal displacements. The observed broadening of the short-distance features in the O-D and O-O $g(r)$ functions are therefore consistent with a distorted structure at the low temperature. Interestingly, at larger distances, the O-O $g(r)$ function at 100 K is somewhat more defined than at 230 K. This implies that despite the local distortions, at larger distances the effects of the lower temperature and hence smaller thermal displacements become visible. Overall, the RMC analysis has shown that the low-temperature diffraction data can be fitted with a fully hydrogen-disordered ice VI model that allows local distortions.

In the next step, the large simulation box was analysed for symmetry as shown in Supplementary Fig. 4. Consistent with expectation, the space group symmetry is $P1$ if no tolerances in the atomic positions are allowed. However, upon increasing the search distance within the FINDSYM software[24], all the previously discussed subgroups of _Pbcn_, which resulted from further distortions, were identified before _Pbcn_ was found above 0.36 Å. In summary, all three approaches including the systematic search for crystallographic subgroups, the analysis of meaningful mechanical distortions, and the symmetry-free RMCProfile refinements point towards _Pbcn_-type structural distortions.

## Discussion

DCl-doped ice VI undergoes _Pbcn_-type distortions at low temperatures and ~1.6 GPa that make sense both from the crystallographic point of view and the mechanics of the crystal structure. These distortions represent an alternative type of phase transition in icy materials in addition to hydrogen ordering at low temperature. While the resulting structure is related to the hydrogen-disordered ice VI, given its lower space group symmetry, it seems justified to assign the Roman numeral XIX to this phase of ice[25]. In line with our earlier work, ice XIX is a deep glassy state of ice VI that has experienced distortions under pressure extensive enough to lower the space group symmetry. In ref. [22] we already suspected that something like this may be possible. Accordingly, the reduction of volume by distorting the hexameric clusters achieves the reduction in entropy of the sample. It is also emphasised that a distorted hydrogen-disordered ice XIX structure with local stress elements is consistent with the Raman[21] and INS data[22] collected earlier for corresponding $H_2O$ samples at ambient pressure. The distortion-induced stress can also be seen from increased widths of the Bragg peaks in X-ray[19,20] and neutron diffraction of recovered $H_2O$ samples[22].

respectively. Close inspection of the O-D $g(r)$ functions shows that the covalent O–D distances at ~0.9 Å are broadened toward lower and higher distances in the 100 K data and the same applies to the hydrogen-bonded O···D distances at ~1.7 Å. The hydrogen-bonded O–O distances at ~2.7 Å in the O-O $g(r)$ are also slightly broadened in the 100 K data. Cooling ice VI to low temperatures

While analysing this data, two related manuscripts have appeared on preprint servers. Firstly, Yamane and co-workers also collected in-situ pressure diffraction data and found very similar trends in the diffraction data at 1.6 GPa and low temperature including the additional Bragg features and the increase in intensity of the ~1.77 Å peak[26]. Consistent with our distortion model, they found that the sample decreases in volume upon the ice VI to ice XIX transition. No crystallographic model was presented and in analogy with previous work[5,6], the additional Bragg features were interpreted in terms of a new hydrogen-ordered structure. To compare the diffraction data collected by us and Yamane and coworkers, the relative intensities of the Bragg peaks at 2.10 and 2.14 Å were determined (see inset in Fig. 2a). The 2.10 Å Bragg peak has similar intensities for both ice VI and XIX, whereas the 2.14 Å peak is absent for ice VI. For the data presented in ref. [26], this ratio is 0.059 compared to 0.197 in our data. This suggests that the pressure in our study must have been somewhat higher leading to more extensive distortions.

Gasser and coworkers recently studied a corresponding recovered sample at ambient pressure and 70 K with neutron diffraction[27]. The peak intensity ratio of their data is 0.062 which may suggest that the decompression has led to partial removal of the distortions. Like Yamane and co-workers, they also interpreted their data in terms of a new hydrogen-ordered structure and suggested $P\bar{4}$ or $Pcc2$ as the space group symmetry. Using the $P\bar{4}$ model, which does not permit full hydrogen order, the average deviation of the fractional occupancies of the hydrogen sites from ½ was 0.16 suggesting weak hydrogen ordering. In contrast to this, we could not observe significant deviations of the occupancies of the hydrogen sites from ½ upon refining the $Pbcn$ model which permits full hydrogen order. The details of this analysis are shown in Supplementary Fig. 5. The average deviation of the fractional occupancies from ½ is estimated to be less than 0.02.

To conclude, ice XIX may be very weakly hydrogen ordered. However, as argued here, the main structural characteristics of ice XIX are the distorted hexameric units of its hydrogen-bonded networks. As evident from the analysis here and also the structural data in ref. [27], the ice $\beta$-XV scenario[20,21], which suggested that ice XIX is more hydrogen-ordered than ice XV, can now be firmly excluded. The appearance of the new Bragg features upon cooling doped ice VI at high pressures has now been firmly established by three independent studies. However, the most intense additional Bragg features were found in our study.

The distortion of subunits in ice under pressure opens up an interesting line of future research. It has been shown that ice VI can be compressed up to 3.8 GPa at 95 K before it transforms to ice VII[28]. It, therefore, remains an open question what happens to ice XIX once compressed to higher pressures. As previously discussed, considering the many different interactions between neighbouring hexameric units in ice XIX arising from the hydrogen disorder, the actual space group symmetry of ice XIX may be best described by $P1$ which is a polar space group. Hence, it seems possible that the different local distortions of the hydrogen-disordered hexameric clusters, which all have different structures at the local level because of the hydrogen disorder, lead to weak ferroelectricity. Such subtle effects may be difficult to pick up in diffraction but they would provide an explanation for the weak ferroelectric behaviour of ice VI found with dielectric spectroscopy at low temperatures[13,14]. If the distortion of hydrogen-disordered phases at low temperatures and high pressures leads to ferroelectric ice in general will need to be investigated in future studies. In this context, it would be interesting to investigate if the hydrogen-disordered ice VII obtained by low-temperature compression[28] is crystallographically fully consistent with the space group symmetry of the corresponding high-temperature phase.

Finally, it is interesting to note that pressure-induced structural distortions of materials containing cluster units are very common. A prime example is the perovskite structure which contains octahedral units linked to one another at the corners[29]. As can be seen in Fig. 1c, the hexameric units of ice VI/XIX can be regarded as distorted octahedra as opposite pairs of the waist oxygen atoms move either up or down along the $c$ direction. $P4/mbm$ octahedral tilting, which is the equivalent of the $Pbcn$ tilting in ice XIX, and the closely related $I4/mcm$ tilting have been observed for a range of perovskites under pressure[30,31]. If other types of distortions of the ice VI structure are possible at even higher pressures will be the focus of future studies.

## Methods

**Neutron diffraction experiment**. A 0.01 molar DCl solution in $D_2O$ was filled into a null-scattering a TiZr gasket and mounted between sintered diamond anvils of a Paris-Edinburgh press on the PEARL beamline[32] at the ISIS Neutron and Muon Source, Didcot, UK. Ice VI was prepared by compression at room temperature to ~1.6 GPa. The temperature of the press was controlled with a reservoir of liquid nitrogen at the bottom of the containment vessel and a heating element. Two long data collection runs were conducted at 230 and 100 K as indicated in Fig. 1a. All neutron diffraction patterns were collected at 90° scattering angle.

**Data analysis**. Rietveld fits were carried out using the GSAS software[33]. For the RMCProfile simulations, $4 \times 4 \times 5$ supercells of the $\sqrt{2} \times \sqrt{2} \times 1$ ice VI unit cell were constructed using the lattice constants obtained from the Rietveld refinements. These structures were fully hydrogen-disordered and harmonic potentials of equal strength were defined between the oxygen and hydrogen sites as well as between the two hydrogen sites along with each hydrogen bond. The equilibrium distances for this were 0.92 and 0.85 Å, respectively. The contributions from the diamond anvils in the diffraction data were subtracted using the calculated profiles from the Rietveld refinement. The background function was also taken from the converged Rietveld refinements.

## Data availability

The data that support the findings of this study are available from the corresponding author upon reasonable request. The crystal structures reported in this study have been deposited at the Cambridge Crystallographic Data Centre (CCDC), under deposition numbers CSD 2080050-2080051.

## Code availability

All software used in this study is freely available and can be accessed as follows: GSAS (https://subversion.xray.aps.anl.gov/trac/EXPGUI), RMCProfile (http://www.rmcprofile.org/) and FINDSYM (https://stokes.byu.edu/iso/findsym.php).

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

## Acknowledgements

We thank the ISIS facility for granting access to the PEARL instrument and Helen Maynard-Caseley for assistance with the experiments. This work was supported by the Engineering and Physical Sciences Research Council through grant EP/E031099/1 and by the Science, Technology, and Facilities Research Council through access to beamtime and other resources.

## Author contributions

J.S.L. conceived the experiment, J.S.L. and C.L.B. conducted the high-pressure neutron diffraction experiment and C.G.S. analysed the diffraction data. A.R.-F. helped putting the new insights into context with previous studies. The manuscript was written with contributions from all authors.

## Competing interests

The authors declare no competing interests.
