## [Peer Review File · Nature Communications]

REVIEWER COMMENTS

Reviewer #1 (Remarks to the Author):

This is the third manuscript in a few weeks on the validation/investigation/analysis of ice XIX... At least, and that is a very important prerequisite, the preprints of the two other papers are properly cited by the authors... so far...

Therefore I shall here focus on the differences to these other papers (fortunately, the chronologically second and this, the third, papers do not claim to be "first"). The deduction of the possible space group (and structure) of ice XIX follows a somehow different approach, considering the expected distortions of the ice VI building blocks (hexameres) leading to a hydrogen-ordered phase with a $\sqrt{2} \times \sqrt{2} \times 1$ ice VI cell. I'd call it a crystal-chemical approach, rather than a group-theoretical one. Tilting, shearing and "squishing" is compatible with a few space groups, only Pbcn is compatible with all three, and the additional Bragg peaks (and intensity variations) in the in situ high pressure powder diffraction pattern of the compressed/cooled/DCl-doped former ice VI.

In a second approach the authors tried a reverse Monte-Carlo fit without symmetry constraint. The presentation of the results is somewhat confusing, but seems to be coherent with the Rietveld approach based on deduced space group symmetries.

What only turns out after longer reading (which may well be something to remedy in a minor revision!), is, that the additional Bragg peaks and the changed Bragg intensities are fitted with one of the distorted structures of ice VI, i.e., the Pbcn model. Yet, it is not the hydrogen ordering which explains the diffraction pattern! The Pbcn model presented here, allows a best fit to the diffraction data, without deriving a lot from the total statistical occupation of the hydrogen positions, ALTHOUGH the model would allow a full hydrogen ordering -- in contrast to one of the models (P-4) suggested (beside Pcc2) by Glasser et al., one of the two preprints on the same subject.

Independent of the fact that I would recommend this paper for publication -- and not only for reasons of scientific "fairness" in this hype around ice XIX -- I would nevertheless encourage the authors (of this paper as well as the recently pre-printed ones) to consider further complementary methods -- notably X-ray diffraction, most obviously -- in order to distinguish a structural distortion under pressure from a hydrogen-ordering (without strong distortion). XRD should be able to provide the necessary complementarity to distinguish the two scenarios that explain the observed neutron diffraction patterns, i.e., it should be "blind" to a pure hydrogen ordering, but not to a structural distortion.

I think the authors should do some (minor) rewriting to make clearer that the models they present consider a distortion in contrast to hydrogen ordering. The paper will appear in the context of the two other ones, altogether they set the scenery, the scientific audience will read all three papers, the readers will be aware of the competition or the complementarity, and the text need to satisfy the expectations. And the expectation of the interested reader is to understand quickly the originality, the difference of approach of each paper.

I recommend this manuscript for publication risking that this is not the last paper on the subject ice XIX, but Nature Communications should be in the first line as well, whenever there is an excited scientific discussion, not to say dispute, present. And for sure, there is or there will be a lot of scientific chitter-chatter in the next months about this new phase of ice, as water and ice is not not a compound like any other, it is possibly the most "important" molecule and solid, as it is perceived by mankind, any news in this field will be "charged with emotions".

Reviewer #2 (Remarks to the Author):

This manuscript reports the measurement and interpretation of neutron scattering measurements of ice (D₂O; prepared as 0.1M DCl solution) at a pressure of 1.6 GPa and at two temperatures: 220 and 100 K. The diffraction pattern measured at 220 K is consistent with that of the hydrogen-disordered phase VI. On cooling to 100 K some new peaks emerge, characteristic of symmetry lowering. The authors account quantitatively for these new peaks by performing Rietveld refinement, and then a slightly unusual RMC refinement against the Bragg scattering profile. The key conclusions drawn are that this lower symmetry phase (labelled ice XIX) (i) has Pbcn symmetry with a unit-cell $\sqrt{2} \times \sqrt{2} \times 1$ with respect to the ice VI cell, (ii) adopts the particular structure shown in Fig. 2b, and (iii) shows no appreciable hydrogen ordering.

I am not a member of the ice community, and can make no comment regarding the importance or otherwise of the characterisation of this ice XIX phase. As a structural chemist I am aware of the significant broader importance of understanding the water phase diagram; however I note that there are a number of existing studies on this phase already in the literature. So I leave judgment on this point to those referees better qualified to comment.

With regards to the structural characterisation itself, there are a number of oddities here in the approach taken, and the extent to which the conclusions drawn are supported by the data presented.

On page 4, the authors note the existence of two weak Bragg features at 2.14 and 2.21 Å in the 100 K measurement. The conclusion is immediately drawn that these imply an increase in the size of the unit cell to a new $\sqrt{2} \times \sqrt{2} \times 1$ supercell. How did the authors exclude the translationengleichen subgroup possibilities? P42/nmc has plenty of such subgroups with different reflection conditions. It would help to identify the indices of the new reflections and use these as a basis for space group determination. I found the narrative on page 5, that explicitly described possible structures inconsistent with the observed reflection conditions, rather odd in this respect. Ultimately it is not clear to me that the authors have identified the highest-symmetry subgroup of phase VI consistent with the additional two reflections. Perhaps they have, but this point needs to be rigorously made.

The refined unit cell implies a statistically significant orthorhombic distortion. Did the authors actually observe any peak splitting? How poor is a fit to the 100 K data using the ice VI model? Is there any hkl-dependent peak broadening evident (I'm conscious here of the difficulty of accurately measuring intrinsic peak widths under high pressure).

If we assume that the data do rigorously support the assignment of the cell and space group as given, then I would have expected the authors to carry out a symmetry mode analysis to identify the key symmetry-breaking structural distortion. Presumably the 'tilt' and 'squish' distortions would emerge naturally, and one would have a magnitude and error on both (and also their covariances). Likewise the distortion associated with hydrogen ordering could be included. Reference is made in the text that the data do not drive any hydrogen ordering. No evidence of this is given, since the hydrogen occupancies in Table 1 are identically 0.5. Given that there are so few additional reflections observed, it is not impossible that the hydrogen ordering distortion shows strong covariance with other distortion modes, and this would need to be ruled in or out. Strictly, all one can say at present is that the data do not require hydrogen ordering, but this is a different point to the conclusion drawn.

The RMC refinement is somewhat unusual in that it only takes into account the Bragg scattering profile and also some O-H and H-H neighbour distance potentials. It is of course unsurprising that the fits obtained are excellent given the large number of degrees of freedom. Since the hydrogen disorder is hardwired into this refinement, it is certainly true that the RMC box shows that hydrogen disordered configurations are capable of fitting the observed data (hardly surprising since the Rietveld analysis has already shown this). However, once again, the inverse cannot be deduced: the authors do not show that hydrogen ordered configurations cannot fit the data. This is especially problematic given the large number of degrees of freedom allowed within the RMC refinement.

Nevertheless, having obtained the RMC-refined model, I could not understand why the authors did not seek to determine the symmetry of its average structure by collapsing onto a suitable subcell. Is it indeed Pbcn, as proposed? And, if so, how similar is this structure solution to that obtained by Rietveld refinement?

So what can we actually say? The Pbcn model is capable of fitting the neutron diffraction pattern at 100 K. That model does not require hydrogen order to fit the observed peak intensities. We have not been given enough evidence to rule out higher-symmetry structural models. Nor can we rule out the consistency of hydrogen-ordered models with the observed data. As things stand, I cannot see that the study can be definitive regarding the nature of the 100 K phase.

Reviewer #3 (Remarks to the Author):

In this work, Salzmann and collaborators use in-situ neutron powder diffraction to study the transformations of ice VI, doped with a small concentration of HCl, upon cooling at $P=1.6$ GPa. It is found that ice VI transforms into ice XIX at low temperatures (about 100 K). The structure of ice VI consists of two interlocking hydrogen-bonded networks with each network being composed of identical "hexameric" units. The authors propose that the structure of ice XIX is similar to that of ice VI but with the hexameric units being tilt periodically throughout the sample.

The manuscript is clear. The results are interesting and supported by the data presented. However, I find the topic of the manuscript to be very specific. I understand that water and ice play a fundamental role in many relevant applications but most of these applications involve low pressures while this study involves ice at >1 GPa. In my view, this article should be published in a different, more specialized, journal (eg, J Phys Chem).

1) The change in the peak of Fig. 2a at $d=1.77$ Å should be shown (as done in the inset of Fig 2a). In addition, it is difficult to compare the scattering patterns from the inset of Fig. 2a and the patterns in Fig. 3b. The authors may want to include the (re-scaled) experimental scattering pattern from the inset of Fig. 2a (blue) into Fig. 3b. Can the authors comment on the agreement/disagreement among the relative heights of the neutron scattering peaks found in the exps (fig 2a) and calculations (fig 3b)?

2) It is mentioned in page 5, line 129, that “The Pnna shearing distortion also shows the new Bragg peaks and will therefore need to be investigated”. I think this issue should be addressed in this work. Otherwise, one is left wondering whether the structure proposed for ice XIX is the correct one.

Minor points: It is mentioned in the abstract that “the new type phase transition found in this work may provide an explanation for previously observed ferroelectric signatures....for other icy materials”. This issue is not discussed in the manuscript.

The revised text in the manuscript is highlighted in yellow.

>> Reviewer #1 (Remarks to the Author):

>> This is the third manuscript in a few weeks on the validation/investigation/analysis of ice XIX... At least, and that is a very important prerequisite, the preprints of the two other papers are properly cited by the authors... so far... Therefore I shall here focus on the differences to these other papers (fortunately, the chronologically second and this, the third, papers do not claim to be "first"). The deduction of the possible space group (and structure) of ice XIX follows a somehow different approach, considering the expected distortions of the ice VI building blocks (hexameres) leading to a hydrogen-ordered phase with a $\sqrt{2} \times \sqrt{2} \times 1$ ice VI cell. I'd call it a crystal-chemical approach, rather than a group-theoretical one. Tilting, shearing and "squishing" is compatible with a few space groups, only Pbcn is compatible with all three, and the additional Bragg peaks (and intensity variations) in the in situ high pressure powder diffraction pattern of the compressed/cooled/DCI-doped former ice VI.

RESPONSE: We thank the reviewer for their comments. In addition to the crystal-chemical approach in the main article, we now present a detailed crystallographic subgroup analysis in the Supplementary Information. All of the previously discussed space groups appear in this new systematic analysis and we carefully tested all candidate structures. Consistent with our previous analysis, the Pbcn space group (including the corresponding Pcnb setting of the same space group) gives the best fit to the data out of all the possible candidate structures from the subgroup analysis. We prefer to keep the crystal-chemical approach in the main article because it is more accessible and refer to the Supplementary Information for the rigorous subgroup analysis. The additional text in the main article starts at line 172 and there is now also a mention of the additional analysis at line 120. Overall, we are delighted that both approaches arrive at the same conclusion.

>> In a second approach the authors tried a reverse Monte-Carlo fit without symmetry constraint. The presentation of the results is somewhat confusing, but seems to be coherent with the Rietveld approach based on deduced space group symmetries.

RESPONSE: We have now improved this section in the manuscript and in the SI we provide an additional detailed symmetry analysis of the large-box structure resulting from the RMC fit in section 4. Consistent with the crystal-chemical approach and the subgroup analysis, this third 'bottom-up' analysis also points towards Pbcn as the space group for ice XIX. The additional text starts at line 223 in the main article. To really emphasise this point, we now have three independent analyses that all pointing towards the same conclusion.

>> What only turns out after longer reading (which may well be something to remedy in a minor revision!), is, that the additional Bragg peaks and the changed Bragg intensities are fitted with one of the distorted structures of ice VI, i.e., the Pbcn model. Yet, it is not the hydrogen ordering which explains the diffraction pattern! The Pbcn model presented here, allows a best fit to the diffraction data, without deriving a lot from the totally statistical occupation of the hydrogen positions, ALTHOUGH the model would allow a full hydrogen ordering -- in contrast to one of the models (P-4) suggested (beside Pcc2) by Glasser et al., one of the two preprints on the same subject.

RESPONSE: Yes, this is the most important point of this manuscript. The diffraction data can be explained using a distorted hydrogen-disordered model. This is also what distinguishes our work from the other two manuscripts. We have now made this point much more strongly in the manuscript including in the abstract. Section 5 in the SI now explores the question of partial hydrogen order in ice XIX in more detail. From the analysis there, we can conclude that the average

absolute deviation of the occupancies of the hydrogen sites from 0.5 is less than 2% and this now mentioned at line 266 in the manuscript. Regarding the distortions themselves, which are a first for ice, the revised manuscript now contains a paragraph on the relationship of the ice VI / XIX distortions with those seen in perovskites starting with line 290. We feel it is important to note that structural distortion under pressure is a very common phenomenon observed for a wide range of materials.

>> Independent of the fact that I would recommend this paper for publication -- and not only for reasons of scientific "fairness" in this hype around ice XIX -- I would nevertheless encourage the authors (of this paper as well as the recently pre-printed ones) to consider further complementary methods -- notably X-ray diffraction, most obviously -- in order to distinguish a structural distortion under pressure from a hydrogen-ordering (without strong distortion). XRD should be able to provide the necessary complementarity to distinguish the two scenarios that explain the observed neutron diffraction patterns, i.e., it should be "blind" to a pure hydrogen ordering, but not to a structural distortion.

RESPONSE: We thank the reviewer for their recommendation to publish our manuscript. Regarding the X-ray diffraction experiments. Firstly, there are already a range of XRD data in the literature in references 19 and 20. These have shown that what we now call ice XIX displays broadened Bragg peaks compared to 'standard' ice VI. These findings support our distortion scenario and we now mention these earlier findings in the manuscript at line 243. Regarding the suggestion that XRD should be "blind" towards hydrogen ordering. This is actually not what we have found in previous studies. Even though X-ray are not significantly scattered by hydrogen atoms, the symmetry changes due to hydrogen ordering can be seen in XRD. Please see Figure 1 in ref. 7 for example. So, if we did some additional XRD measurements and saw a symmetry change, we would NOT be able to tell if it arises from hydrogen order or from structural distortions. In-situ neutron diffraction, as used in this manuscript, is really the gold standard since it enables us to distinguish between the two possibilities. In this context, we note that we have now include a detailed analysis of the level of hydrogen order in ice XIX as mentioned earlier.

>> I think the authors should do some (minor) rewriting to make clearer that the models they present consider a distortion in contrast to hydrogen ordering. The paper will appear in the context of the two other ones, altogether they set the scenery, the scientific audience will read all three papers, the readers will be aware of the competition or the complementarity, and the text need to satisfy the expectations. And the expectation of the interested reader is to understand quickly the originality, the difference of approach of each paper.

RESPONSE: We thank the reviewer for this comment. This has now been made much clearer in the abstract and throughout the manuscript.

>> I recommend this manuscript for publication risking that this is not the last paper on the subject ice XIX, but Nature Communications should be in the first line as well, whenever there is an excited scientific discussion, not to say dispute, present. And for sure, there is or there will be a lot of scientific chitter-chatter in the next months about this new phase of ice, as water and ice is not not a compound like any other, it is possibly the most "important" molecule and solid, as it is perceived by mankind, any news in this field will be "charged with emotions".

RESPONSE: We thank the reviewer again for their recommendation to publish our manuscript and we definitively share the excitement about the new discovery.

>> Reviewer #2 (Remarks to the Author):

>> This manuscript reports the measurement and interpretation of neutron scattering measurements of ice (D₂O; prepared as 0.1M DCl solution) at a pressure of 1.6 GPa and at two temperatures: 220 and 100 K. The diffraction pattern measured at 220 K is consistent with that of the hydrogen-disordered phase VI. On cooling to 100 K some new peaks emerge, characteristic of symmetry lowering. The authors account quantitatively for these new peaks by performing Rietveld refinement, and then a slightly unusual RMC refinement against the Bragg scattering profile. The key conclusions drawn are that this lower symmetry phase (labelled ice XIX) (i) has Pbcn symmetry with a unit-cell $\sqrt{2} \times \sqrt{2} \times 1$ with respect to the ice VI cell, (ii) adopts the particular structure shown in Fig. 2b, and (iii) shows no appreciable hydrogen ordering.

RESPONSE: We thank the reviewer for this accurate summary.

>> I am not a member of the ice community, and can make no comment regarding the importance or otherwise of the characterisation of this ice XIX phase. As a structural chemist I am aware of the significant broader importance of understanding the water phase diagram; however I note that there are a number of existing studies on this phase already in the literature. So I leave judgment on this point to those referees better qualified to comment.

RESPONSE: With respect to the other studies in the literature, we would like to point out that our study puts forward a different structural model to explain the crystallographic changes. While distortions of subunits are well known for other materials such as perovskites, this is a first for ice.

>> With regards to the structural characterisation itself, there are a number of oddities here in the approach taken, and the extent to which the conclusions drawn are supported by the data presented. On page 4, the authors note the existence of two weak Bragg features at 2.14 and 2.21 Å in the 100 K measurement. The conclusion is immediately drawn that these imply an increase in the size of the unit cell to a new $\sqrt{2} \times \sqrt{2} \times 1$ supercell. How did the authors exclude the translationsgleichen subgroup possibilities? P42/nmc has plenty of such subgroups with different reflection conditions. It would help to identify the indices of the new reflections and use these as a basis for space group determination.

RESPONSE: We thank the reviewer for this comment and agree that more information is needed. The Supplementary Information now contains five sections that provide additional information on the crystallography of the problem. Firstly, in section 1 we fitted the ice VI crystallographic model to the low-temperature diffraction pattern. Figure S1 also includes the tickmarks of the P1 structure with the ice VI lattice constants. As can be seen, the two new peaks are not possible in P1 which means that the size of the unit cell needs to increase. Indexing suggests the new peaks as 231/321 and 212, respectively, of a $\sqrt{2} \times \sqrt{2} \times 1$ supercell. In addition to this, we have now carried out a detailed subgroup analysis of P42/nmc as described in section 2 of the SI. Out of all the translationsgleichen subgroups, Ccca is the only one that is not zellengleich. The transition from P42/nmc to Ccca increases the volume of the unit cell by $\sqrt{2} \times \sqrt{2} \times 1$ which is exactly what we need. However, the reflection conditions of Ccca still do not allow the new Bragg peak despite the larger cell. We therefore systematically explored all subgroups of the translationsgleichen subgroups of P42/nmc including Ccca. The results are summarised in Table S1. Reassuringly, all space groups derived using our crystal-chemistry approach in the main article appear in this analysis. They are marked with asterisks in Table S1. Based on the subgroup analysis, we arrive at six possible candidate space groups. Please note that the settings of the resulting space groups may vary in the two different approaches. For example, Pcnb and Pbcn are two different settings of space group 60 but in our case they describe the same structure with different definitions of the

axes. In section 3, we then test the six candidate structures by Rietveld refinements against the experimental data. Consistent with the analysis in the main article, Pbcn/Pcnb gives the best fit to the data. This means that space group 60 results from both our crystal-chemistry approach considering the possible meaningful distortions as well as from a systematic crystallographic subgroup analysis. In this context, we have now also more thoroughly addressed the question of the Pnna model which was previously discussed in the main article. Figure S3 shows that the fit using Pnna is not as good as for Pbcn/Pcnb. In addition to the new sections in the SI, the text in the main article has been modified at lines 109, 119 and 172.

>> I found the narrative on page 5, that explicitly described possible structures inconsistent with the observed reflection conditions, rather odd in this respect. Ultimately it is not clear to me that the authors have identified the highest-symmetry subgroup of phase VI consistent with the additional two reflections. Perhaps they have, but this point needs to be rigorously made.

RESPONSE: The approach for finding the space group in the main article is by considering the various types of possible distortions. This then of course produces models that are inconsistent with the observed reflections, but they can simply be excluded. We thought that this approach is more accessible for a general readership compared to the crystallographic subgroup analysis. If the reviewer agrees, we would prefer to keep this approach in the main article and refer the reader to the SI for the more systematic subgroup analysis. Modified parts in the manuscript are at lines 120 & 172. Just to mention this point again, both approaches arrive at the same conclusion and all space groups suggested in the main article appear in the subgroup analysis in the SI.

>> The refined unit cell implies a statistically significant orthorhombic distortion. Did the authors actually observe any peak splitting? How poor is a fit to the 100 K data using the ice VI model? Is there any hkl-dependent peak broadening evident (I'm conscious here of the difficulty of accurately measuring intrinsic peak widths under high pressure).

RESPONSE: The simple answer is that we do not see any, nor would we expect to since the orthorhombic distortion is ~0.2 % and our resolution is 0.7%. However, as we have seen, the requirement that ice XIX has the same network topology as ice VI implies an orthorhombic symmetry whether or not a and b have equal length and so the question as to whether they are different is not important to the issue of the symmetry. In any case, as the reviewer notes the orthorhombic distortion is well determined (in the refinements it is a ~80 sigma effect). As mentioned earlier, we now show the fit of the ice VI model to the low-temperature data in Figure S1. The fit is not terrible. But it fails to reproduce the new additional peaks and the intensity of the 1.77 Å peak. This means that the structures are closely related but a lower symmetry is needed to fit the low-temperature data which we have explored in detail. We could not observe any systematic hkl-dependent peak broadening.

>> If we assume that the data do rigorously support the assignment of the cell and space group as given, then I would have expected the authors to carry out a symmetry mode analysis to identify the key symmetry-breaking structural distortion. Presumably the 'tilt' and 'squish' distortions would emerge naturally, and one would have a magnitude and error on both (and also their covariances). Likewise the distortion associated with hydrogen ordering could be included. Reference is made in the text that the data do not drive any hydrogen ordering. No evidence of this is given, since the hydrogen occupancies in Table 1 are identically 0.5. Given that there are so few additional reflections observed, it is not impossible that the hydrogen ordering distortion shows strong covariance with other distortion modes, and this would need to be ruled in or out. Strictly, all one can say at present is that the data do not require hydrogen ordering, but this is a different point to the conclusion drawn.

RESPONSE: The question of hydrogen order in our sample is now addressed in section 5 of the SI and a reference to this is given in line 266 in the main article. The results from a free refinement of the Pbcn model including the fractional occupancies of the hydrogen sites are shown in Figure S5. The ice rules were implemented with linear constraints and composition restraints in GSAS. To get an estimate for the errors of the occupancies, we also refined the Pbcn model against the ice VI data. The deviations of the occupancies from 0.5 are quite small. Based on these refinements, we estimate the average absolute deviation from 0.5 to be less than 0.02 for ice XIX. This illustrates that ice XIX is only weakly hydrogen ordered if at all. Regarding a symmetry mode analysis, we would argue that our analysis of the mechanically meaningful distortions in the main article goes into this direction. In particular since we have now also established that all meaningful distortions are described by subgroups of Ccca which is the only translationsgleiche subgroup of P4₂/nmc that requires the larger cell. In this context, it is interesting to mention that Ccca does not permit any hydrogen order. However, it can already describe the tilting distortion. This means that the initial symmetry breaking is caused by distortions and not by hydrogen ordering. These points are now discussed in section 5 of the SI.

>> The RMC refinement is somewhat unusual in that it only takes into account the Bragg scattering profile and also some O-H and H-H neighbour distance potentials. It is of course unsurprising that the fits obtained are excellent given the large number of degrees of freedom. Since the hydrogen disorder is hardwired into this refinement, it is certainly true that the RMC box shows that hydrogen disordered configurations are capable of fitting the observed data (hardly surprising since the Rietveld analysis has already shown this). However, once again, the inverse cannot be deduced: the authors do not show that hydrogen ordered configurations cannot fit the data. This is especially problematic given the large number of degrees of freedom allowed within the RMC refinement.

RESPONSE: We fully agree with the reviewer that the RMC analysis is somewhat unusual. The Rietveld analysis paired with the two independent searches for subgroups stand for themselves and our analysis could have perhaps stopped there. However, ice is a complex material because of the orientational disorder of the molecules which leads to small positional disorder of the oxygen atoms. In case of ice VI / XIX, because of the orientational disorder a vast range of possible structures exists for the hexameric units going through all the possible permutations of the orientations. Considering the interactions with neighbouring units, which become important at high pressures, astronomically large numbers of different environments can be expected. All of this is not considered in a Rietveld refinement where the average structure is optimised. Because of this speciality for ice, we decided to carry out a 'bottom-up' approach for solving the structure of ice XIX by using a large-box model of ice VI. The problem with the large-box approach is that while complete disorder is easily implemented, there are large numbers of different ordered structures. The situation is further complicated by the fact that the structures would not necessarily need to be fully ordered but could be partially ordered. So producing a set of possible start structures would be a very difficult if not impossible task. The question of hydrogen order is certainly more easily addressed with the Rietveld analysis that we have carried out. Nevertheless, the $g(r)$ data shown in Figure 4b gives some interesting structural insights: the ice XIX structure is locally more distorted but more ordered at larger distances compared to ice VI. If the reviewer agrees we would therefore like to keep this analysis.

>> Nevertheless, having obtained the RMC-refined model, I could not understand why the authors did not seek to determine the symmetry of its average structure by collapsing onto a suitable subcell. Is it indeed Pbcn, as proposed? And, if so, how similar is this structure solution to that obtained by Rietveld refinement?

RESPONSE: Yes, this is a valid point. Section 4 in the SI now deals with the symmetry analysis of the large RMC simulation box. As shown in Figure S4, Pbcn is indeed identified for search distances above 0.36 Å. The sequence of space groups observed at smaller search distances is the same as we have discussed regarding the subgroups of Pbcn in the main article (line 185). In conclusion, we can say that this analysis also points towards Pbcn as the key space group in this system. Of course, the RMC approach is prone to deliver a more disordered structure compared to the Rietveld analysis. For this reason, we believe that the best way to analyse the RMC structure is the analysis shown in Figure 4b where we are comparing the results from ice VI and ice XIX. A reference to section 4 in the SI is now given starting with line 223 in the main article.

>> So what can we actually say? The Pbcn model is capable of fitting the neutron diffraction pattern at 100 K. That model does not require hydrogen order to fit the observed peak intensities. We have not been given enough evidence to rule out higher-symmetry structural models. Nor can we rule out the consistency of hydrogen-ordered models with the observed data. As things stand, I cannot see that the study can be definitive regarding the nature of the 100 K phase.

RESPONSE: We thank the reviewer for the constructive comments which have certainly helped us improve our manuscript. We believe that the various additional analyses described earlier now firmly address the questions of higher-symmetry models and hydrogen order.

>> Reviewer #3 (Remarks to the Author):

>> In this work, Salzmann and collaborators use in-situ neutron powder diffraction to study the transformations of ice VI, doped with a small concentration of HCl, upon cooling at P=1.6 GPa. It is found that ice VI transforms into ice XIX at low temperatures (about 100 K). The structure of ice VI consists of two interlocking hydrogen-bonded networks with each network being composed of identical “hexameric” units. The authors propose that the structure of ice XIX is similar to that of ice VI but with the hexameric units being tilt periodically throughout the sample.

RESPONSE: We thank the reviewer for this accurate summary.

>> The manuscript is clear. The results are interesting and supported by the data presented. However, I find the topic of the manuscript to be very specific. I understand that water and ice play a fundamental role in many relevant applications but most of these applications involve low pressures while this study involves ice at >1 GPa. In my view, this article should be published in a different, more specialized, journal (eg, J Phys Chem).

RESPONSE: We thank the reviewer for the assessment that the manuscript is clear and that the results are interesting and supported by the data. Regarding the publication in a more specialised journal, we would like to point out the novelty aspects of this work. For the first time, a new phase of ice is reported that forms through distortions of clusters within the structure. As described in the revised version of the manuscript, such distortions under pressure are well-known for other materials such as perovskites (line 290) but not for ice. This is in our opinion an exciting discovery and, as explained in the manuscript, the structural distortions may be origin for weak ferroelectricity in ice which will need to be explored in greater detail in the future.

>> 1) The change in the peak of Fig. 2a at d=1.77 Å should be shown (as done in the inset of Fig 2a). In addition, it is difficult to compare the scattering patterns from the inset of Fig. 2a and the patterns in Fig. 3b. The authors may want to include the (re-scaled) experimental scattering pattern from the inset of Fig. 2a (blue) into Fig. 3b. Can the authors comment on the

agreement/disagreement among the relative heights of the neutron scattering peaks found in the exps (fig 2a) and calculations (fig 3b)?

RESPONSE: The region of the important Bragg peak at 1.77 Å is now shown as insets in Figure 2a and 4a. Figure 3b shows how the crystallographic features change when the hexameric units are tilted. However, at this stage, the “squishing” modes are not included and they are really needed to obtain the best possible fit as explained later in the manuscript. For this reason, we would like to refrain from showing the experimental data in Figure 3b. The purpose of this figure is really to illustrate that the crystallographic model responds in the correct way when structural distortions are included. However, for a complete fit of the experimental data, the Rietveld approach is needed which is shown in Figure 2a.

>> 2) It is mentioned in page 5, line 129, that “The Pnna shearing distortion also shows the new Bragg peaks and will therefore need to be investigated”. I think this issue should be addressed in this work. Otherwise, one is left wondering whether the structure proposed for ice XIX is the correct one.

RESPONSE: We thank the reviewer for pointing this out. We now provide a complete and systematic subgroup analysis of the ice VI space group in the Supplementary Information. To our delight, all of the space groups derived on the basis of crystal-chemical considerations in Figure 3a including the Pnna structure appeared in this analysis. Please see Table S1 and Figure S2. All possible candidate structures were then tested against the experimental data, again including Pnna, and it was found that the Pbcn structure gives the best fit to the data as shown in Figure S3. Please note that for technical reasons we obtained Pcnb from the subgroup analysis which is, however, equivalent to Pbcn as explained in the SI. To make clear that the Pnna model did not give the best fit to the data, a sentence was added in line 176 in the main article.

>> Minor points: It is mentioned in the abstract that “the new type phase transition found in this work may provide an explanation for previously observed ferroelectric signatures....for other icy materials”. This issue is not discussed in the manuscript.

RESPONSE: We thank the reviewer for pointing this out. There was a brief discussion about this in the original manuscript. However, we have now extended this and the entire paragraph starting with line 275 now deals with the question of ferroelectricity and if the phenomenon can be observed for other icy materials as well.

REVIEWERS' COMMENTS

Reviewer #1 (Remarks to the Author):

The authors considered carefully all comments made by all reviewers, not only mine, and what is even more important to me, personally, is, that the responses to my remarks confirmed basically, not only that my remarks were somehow "pertinent", but more importantly, that I understood the original manuscript properly, which confirms my original "judgment" that an only "minor revision" would be adequate.

It is now much clearer to the reader, that the notable result of this investigation differs substantially from the two previous papers on ice XIX, which does NOT appear to be a second hydrogen-ordered low-temperature phase of ice VI like ice XV.

I appreciate the detailed rebuttal letter and obviously recommend the manuscript for publication in Nature Communications.

Reviewer #2 (Remarks to the Author):

The authors have addressed all substantive points raised and have done a good job of improving this manuscript. I am happy to recommend acceptance in Nature Communications and publication as is.

Reviewer #3 (Remarks to the Author):

I appreciate the additional results added in the Supplementary Information (SI) of the manuscript. Specifically, in the new SI, the authors fit their neutron diffraction data using various suitable crystallographic structures. They show that the best fit to the ice XIX data is obtained with the Pbcn space group, as they originally proposed. In addition, they show that the previously discussed Pnna model should be discarded.

A few changes have also been added to the text. I cannot point to what specific parts of the text have been improved but I can now appreciate the message that the authors tried to convey in their first submission, i.e., that the proposed molecular-level mechanism for the ice VI-ice XIX phase transition has never been observed/proposed in low-temperature ices, and that this mechanism contrasts the usual phase transitions between orientational disordered-ordered ices reported in the past. In addition, the manuscript also proposes that the structure of ice XIX presented here could provide an explanation for the weak ferroelectric behavior of ice VI found in previous studies.

The manuscript is still specific, but I can now grasp the novelty of the ideas presented in this work. Accordingly, I support the manuscript for publication.

Reviewer #1 (Remarks to the Author):

The authors considered carefully all comments made by all reviewers, not only mine, and what is even more important to me, personally, is, that the responses to my remarks confirmed basically, not only that my remarks were somehow "pertinent", but more importantly, that I understood the original manuscript properly, which confirms my original "judgment" that an only "minor revision" would be adequate.

RESPONSE: We are delighted to hear that the reviewer thinks that we have carefully considered the comments from all reviewers.

It is now much clearer to the reader, that the notable result of this investigation differs substantially from the two previous papers on ice XIX, which does NOT appear to be a second hydrogen-ordered low-temperature phase of ice VI like ice XV.

RESPONSE: We thank the reviewer for asking us to emphasise this important difference with respect to the other two papers.

I appreciate the detailed rebuttal letter and obviously recommend the manuscript for publication in Nature Communications.

RESPONSE: We thank the reviewer for their recommendation to publish our work in Nature Communications.

Reviewer #2 (Remarks to the Author):

The authors have addressed all substantive points raised and have done a good job of improving this manuscript. I am happy to recommend acceptance in Nature Communications and publication as is.

RESPONSE: We thank the reviewer for their recommendation to publish the article as is.

Reviewer #3 (Remarks to the Author):

I appreciate the additional results added in the Supplementary Information (SI) of the manuscript. Specifically, in the new SI, the authors fit their neutron diffraction data using various suitable crystallographic structures. They show that the best fit to the ice XIX data is obtained with the Pbcn space group, as they originally proposed. In addition, they show that the previously discussed Pnna model should be discarded.

RESPONSE: We thank the reviewer again for their constructive comments which have led us to expand the SI.

A few changes have also been added to the text. I cannot point to what specific parts of the text have been improved but I can now appreciate the message that the authors tried to convey in their first submission, i.e., that the proposed molecular-level mechanism for the ice VI-ice XIX phase transition has never been observed/proposed in low-temperature ices, and that this mechanism contrasts the usual phase transitions between orientational disordered-ordered ices reported in the past. In addition, the manuscript also proposes that the structure of ice XIX presented here could provide an explanation for the weak ferroelectric behavior of ice VI found in previous studies.

The manuscript is still specific, but I can now grasp the novelty of the ideas presented in this work. Accordingly, I support the manuscript for publication.

RESPONSE: We thank the reviewer for their support to publish our manuscript.